# Transparent Cross-Flow Platform as Chemiluminescence Detection Cell in Cross Injection Analysis

**DOI:** 10.3390/molecules28031316

**Published:** 2023-01-30

**Authors:** Thachkorn Somboonsuk, Phoonthawee Saetear, Thitirat Mantim, Nuanlaor Ratanawimarnwong, Prapin Wilairat, Nathawut Choengchan, Duangjai Nacapricha

**Affiliations:** 1Flow Innovation-Research for Science and Technology Laboratories (Firstlabs), Ratchathewi District, Bangkok 10110, Thailand; 2Department of Chemistry and Center of Excellence for Innovation in Chemistry, Faculty of Science, Mahidol University, Rama 6 Road, Ratchathewi District, Bangkok 10400, Thailand; 3Department of Chemistry, Faculty of Science, Srinakharinwirot University, Sukhumvit 23, Watthana District, Bangkok 10110, Thailand; 4Department of Chemistry and the Applied Analytical Chemistry Research Unit, Faculty of Science, King Mongkut’s Institute of Technology Ladkrabang, Ladkrabang District, Bangkok 10520, Thailand

**Keywords:** cross injection analysis, chemiluminescence, cross-flow, cobalt, paracetamol

## Abstract

This work presents the use of a transparent ‘Cross Injection Analysis’ (CIA) platform as a flow system for chemiluminescence (CL) measurements. The CL-CIA flow device incorporates introduction channels for samples and reagents, and the reaction and detection channels are in one acrylic unit. A photomultiplier tube placed above the reaction channel detects the emitted luminescence. The system was applied to the analysis of (i) Co(II) via the Co(II)-catalyzed H_2_O_2_-luminol reaction and (ii) paracetamol via its inhibitory effect on the catalytic activity of Fe(CN)_6_^3−^ on the H_2_O_2_-luminol reaction. A linear calibration was obtained for Co(II) in the range of 0.002 to 0.025 mg L^−1^ Co(II) (r^2^ = 0.9977) for the determination of Co(II) in water samples. The linear calibration obtained for the paracetamol was 10 to 200 mg L^−1^ (r^2^ = 0.9906) for the determination of pharmaceutical products. The sample throughput was 60 samples h^−1^. The precision was ≤4.2% RSD. The consumption of the samples and reagents was ca. 170 µL per analysis cycle.

## 1. Introduction

Chemiluminescence (CL) is a phenomenon whereby light is emitted from molecules, with an electronic excited state resulting from a chemical reaction [1]. The intensity of the emitted light is dependent on the number of excited molecules, and, thus, measurements of the emitted light are suitable for quantitation work. CL has been widely used in various applications in chemical analyses [2,3,4,5,6,7,8] due to its highly selective and sensitive features. Luminescence detection only requires a photomultiplier tube (PMT) or a photodiode installed in an appropriate light-tight container [3,9].

There are various CL reactions that have been employed for chemical analyses [2,4,5,6,7,8,10,11]. However, the reaction between 5-amino-2,3-dihydrophthalazine-1,4-dione (luminol) and hydrogen peroxide is the most used process [2,7,10]. Transition metal cations (Co(II), Cu(II), Cr(II), Fe(II), Fe(III), Hg(II), Mn(IV) and Ni(II)) and their complexes (e.g., ferrocene and ferricyanide) are the catalysts of this reaction [7]. This catalytic property can thus be used to quantify trace amounts of these metals or complexes [12,13,14,15,16]. Likewise, the suppression of CL by certain molecules can also be used to quantify compounds [17,18,19,20]. The applications of CL in quantitative analyses are extensive and well-documented [2,3,4,5,6,7,8,9,10,11,12,13,14,15,16,17,18,19,20]. There are review articles that provide information on CL applications that use various types of samples, such as environmental [5], wastewater [4], food [3], pharmaceuticals [6], and clinical and forensic [10] samples. Progress and advances in technical developments can be found in two recent reviews focusing on biosensors [3] and flow analyses [2].

CL emission is a transient process. Apart from the chemical factors, the intensity of CL is influenced by physical conditions. To achieve a high precision, analyses using CL are best carried out in an automated flow system for the reproducible control of liquid introduction and mixing, as well as ensuring the exact timing of the light detection. To obtain the highest detection sensitivity, the point of introduction of the sample/reagents and the flow-through detection cell is usually set in close proximity to the detector [2].

CL flow cells are made of silica, glass or transparent/translucent polymeric materials, and they are commercially available with limited designs. Many researchers have therefore designed and fabricated their own flow cells [2,11,21,22,23,24,25,26,27,28,29,30,31,32,33,34,35,36]. Some flow cells have one flow path for the detection of CL light, with one inlet and one outlet [28,29,30,31,32,33,34,35,36]. The mixing of the sample and CL reagent streams occurs before the solution enters the detection cell. The formation of excited-state molecules is very prompt (occurring within the mixing time), which is followed by an exponential decay of the emitted light [2]. Thus, the maximal level of light is not detected. Other flow cell designs have inlet ports located inside the detection chamber for the introduction of both samples and reagents [21,22,23,24,25,26]. There are mixing designs for rapid mixing, such as (i) magnetic stirring [21], (ii) vortex mixing [22] and (iii) swirling mixing [23]. Spiral [24,25,26], sinusoidal [25] and serpentine [25,26] detection flow lines have been employed. In the lab-on-valve (LOV) manifold, a PMT is mounted on top of a transparent LOV platform to directly detect the CL emitted from the reaction area of the platform [27]. The design of the lab-on-valve (LOV) flow conduit is versatile for use with various chemical analyses (not only for chemiluminescence). There are at least four built-in inlet ports for the introduction and mixing of samples and reagents. Due to the transparency of the LOV platform, there is no need to modify the flow manifold for CL detection.

In this work, we present the utilization of a transparent fluidic ‘Cross Injection Analysis’ (CIA) platform [37,38] as a CL flow cell. CIA employs peristaltic pumps and flexible tubing for liquid handling, and it is capable of maintaining a constant pressure without the need for injection valves. The CIA unit is easily produced with a simple milling procedure. The narrow liquid conduit is constructed by drilling out a transparent rectangular acrylic slab to create multiple straight channels. The CIA platform comprises one main channel (denoted as the x-channel) along the length of the acrylic slab and multiple perpendicular channels (denoted as the y-channels) along the width of the slab. The x-channel is filled with the carrier solution, and the y-channels are filled with the respective sample or reagent solutions. The intersections of the x- and y-channels on the CIA platform are thus initially filled with the sample solution or the respective CL reagents. With flow only in the x-flow channel, the various liquid zones filling the intersections of the x–y channels are pushed to mix together along the carrier (x-channel) conduit by the turbulent flow [37]. CL emission is produced in the final mixed zone, and this is detected by a side-on PMT, which is situated above the detection area of the CIA unit. The CIA platform has two functions; i.e., it functions as a flow reactor and as a detection flow chamber. Two CIA systems are demonstrated, i.e., (i) the determination of Co(II) based on the Co(II)-catalyzed H_2_O_2_-luminol reaction and (ii) an analysis of paracetamol via its inhibitory effect of the catalytic activity of Fe(CN)_6_^3−^ on the H_2_O_2_-luminol reaction. There are some publications reporting on the development of flow-based systems for the quantitative analyses of Co(II) [22] and paracetamol [17] using the oxidation of luminol as a CL reaction. However, these systems use a valve for the injection of either the sample [22] or the reagent [17]. Furthermore, extra detection flow cells are required. In contrast, our CIA platform has neither valves nor detection cells. The developed CIA systems are also applied to water samples (for the Co(II) determination) and pharmaceutical products (for the paracetamol analysis) in order to demonstrate their applications in real samples. To the best of our knowledge, this is the first report on the use of a transparent CIA unit as both the analytical platform and the flow-through CL detection cell for the determination of Co(II) and paracetamol.

## 2. Results and Discussion

### 2.1. Preliminary Tests of Light Detection with the CIA Platform

Experiments were first carried out to determine the feasibility of using a transparent acrylic material as a light detection cell. The CIA manifold displayed in Appendix A used for the Co(II)-catalyzed H_2_O_2_-luminol reaction was operated by using the CIA control box and by employing the operating sequence given in Appendix A. In an adequately darkened room, it was visually observed that CL light was produced inside the CIA conduit. An increased light intensity was observed when the Co(II) concentration was increased from 0.01 to 0.5 µmol L^−1^. To record the light intensity, the lens of a digital camera was inserted into the circular hole at the top of the light-tight box (as shown in Figure 1a), and an image of the chemiluminescence was recorded. The concentrations of the Co(II) standards were increased in the range of 0.1–10 mmol L^−1^. Figure 1b shows an example of the recorded image of the CL light emission along the CIA conduit. A quantitative analysis of the images was carried out using the ImageJ program. Appendix A shows that the gray and blue intensity values of the images increased with the increase in the concentration of Co(II). These results show that the CIA platform can be used as a flow cell for quantitative CL measurements. In addition, a spiral flow cell was installed at the exit of the x-axis channel, and the light intensity was recorded. However, a much lower light intensity was observed with the spiral flow cell. Therefore, the optimal position for detecting the CL emission is at the cross-flow position of the CIA platform.

### 2.2. Optimization of CIA System I for Co(II) Analysis

The setup of the CIA system for chemiluminescence detection was carried out as described in Section 3.1. The platform was connected to the tubing, as described in Section 3.2, with a photomultiplier used as the detector. The operating steps are described in Section 3.2. Figure 2a shows examples of four signal profiles obtained from the H_2_O_2_-luminol reaction catalyzed by Co(II). As expected, the signal increased with the increase in the concentration of the Co(II) catalyst from 0.00 to 0.03 mg L^−1^ Co(II) (the profile of zero Co(II) (reagent blank) was used as the baseline signal). The measured CL intensity achieved the maximum intensity within the mixing time of the CIA conduit (Step 2 in the routine procedure in Appendix A). The CL signal finally fell to the baseline level as the conduit flowed with more H_2_O_2_ solution along y-axis channels 3 and 4 (Step 3 in the routine procedure in Appendix A). It was observed that there was always a spike at the beginning of this rinsing process (at ca. 40 s in Figure 2a). This was due to the H_2_O_2_ in channels 3 and 4 reacting with residual luminol in the x-axis channel.

The concentrations of the luminol and H_2_O_2_ reagents were optimized. For the luminol reagent, the sensitivity increased with an increase in the luminol concentrations. In order to achieve the highest sensitivity, the highest studied luminol concentration (2.5 mmol L^−1^) was selected. For the H_2_O_2_ reagent, the sensitivity did not change significantly within the investigated concentration range of 0.25–2.0% (*w*/*w*). For this work, the H_2_O_2_ reagent concentration of 0.5% *w*/*w* was chosen. The data of this optimization study are shown in Appendix A.

A linear relationship between the maximum height of the CI intensity and the Co(II) concentration was observed for 0.002 to 0.025 mg L^−1^ Co(II), (signal (V) = (69.3 ± 2.34)·(Co(II), mg L^−1^) + (2.50 ± 0.03): r^2^ = 0.9977). An example of the calibration plot of the standard Co(II) is presented in Appendix A. These results demonstrate that the CIA platform can be used as the CL flow system for the quantitative analysis of Co(II).

### 2.3. CIA System II for Paracetamol Analysis

The CIA System II platform was connected to the tubing as described in Section 3.3, with a photomultiplier used as the detector (see Section 3.1). The operating steps are described in Section 3.3. Figure 2b shows a series of recorded CL profiles obtained from the Fe(CN)_6_^3−^-catalyzed H_2_O_2_-luminol reaction, with the inhibitory effect of paracetamol. Similar to the Co(II)-CIA system, the maximum peak signals were observed during the mixing step (Step 2 of the routine procedure in Appendix A). The paracetamol reacted with Fe(CN)_6_^3−^, leading to a significant decrease in the light emitted from the Fe(CN)_6_^3−^-catalyzed H_2_O_2_-luminol reaction. There were no spike peaks observed, as shown in Figure 2b, during the rinsing step. In the presence of the paracetamol inhibitor, the CL intensity decayed completely within Step 2 (the routine procedure) due to the longer mixing time of 30 s (cf. 20 s for the Co(II) analysis).

The concentrations of the luminol, H_2_O_2_ and Fe(CN)_6_^3−^ were optimized. Comparable sensitivities were obtained with all tested concentrations of the luminol reagent (3–5 mmol L^−1^, Appendix A). We therefore selected the lowest luminol concentration of 3 mmol L^−1^. The concentration of the H_2_O_2_ reagent was investigated from 0.5 to 2.0% *w*/*w* (see Appendix A). The sensitivity sharply increased for the concentrations from 0.5 to 1.0% *w*/*w*, but it remained almost constant up to 2.0% *w*/*w*. To avoid the formation of bubbles, 1.0% *w*/*w* H_2_O_2_ was selected. The K_3_Fe(CN)_6_ reagent reacted with the paracetamol, and the remaining K_3_Fe(CN)_6_ acted as a catalyst for the H_2_O_2_-luminol reaction. The maximum concentration of the paracetamol in this study was 1.0 mmol L^−1^. Thus, for a 1:1 reaction between paracetamol and Fe(CN)_6_^3−^, the concentration of the K_3_Fe(CN)_6_ reactant must be higher than 1.0 mmol L^−1^. As expected, the sensitivity of the analysis increased when the concentration of K_3_Fe(CN)_6_ was increased from 1.0 to 2.5 mmol L^−1^ (see Appendix A). However, the inner filter effect of the luminol emission at 425 nm by the K_3_Fe(CN)_6_ solution resulted in a decrease in the sensitivity of concentrations greater than 2.5 mmol L^−1^ (see Appendix A). Thus, 2.5 mmol L^−1^ K_3_Fe(CN)_6_ was chosen as the optimal concentration of this reagent. We obtained a linear relationship between the percent decrease in the maximum intensity (% decrease) and the concentration of paracetamol from 10 to 200 mg L^−1^ paracetamol (% decrease = (0.35 ± 0.02) · (paracetamol, mg L^−1^) + (6 ± 2): r^2^ = 0.9906) (see Section 2.4 for the calculation of % decrease). An illustration of the calibration curve for the standard paracetamol is depicted in Appendix A.

### 2.4. Analytical Performance of the CL-CIA Systems

Using the optimized conditions, the CL-CIA methods developed for both Co(II) and paracetamol were evaluated for their performances. Table 1 provides a summary of the performances of the two systems. The linear calibration of Co(II) is a plot between the maximum PMT signal (volt) against the concentration of Co(II). For paracetamol, the calibration is a plot of the percent decrease in the maximum PMT signal against the concentration of paracetamol, using (I_0_ − I)/I_0_ × 100, where I_0_ and I are the maximum PMT signals in the absence and in the presence of paracetamol, respectively. The results in Table 1 show that both CL-CIA systems gave a good linearity (r^2^ of 0.99), with good precisions (%RSD ≤ 4.2%) and recoveries (87–110%). Similar to other flow-injection-based methods, these CL-CIA methods provide a high sample throughput of 60 samples h^−1^. The systems require very low volumes of reagents and samples, using ca. 170 µL per cycle.

### 2.5. Applications and Validation

The CL-CIA system developed for the determination of Co(II) was applied to analyze five water samples. Flame atomic absorption spectrometry (FAAS) was used as the reference method. The results show that the levels of Co(II) in all water samples were below the LOQ. In order to validate the method, standard Co(II) was spiked into these water samples to give a concentration of 1.00 mg L^−1^. The spiked water samples were diluted 50-fold (0.020 mg L^−1^) before carrying out analyses using the developed CL-CIA. The FAAS method did not require any dilution of the spiked water samples. The results in Figure 3a show a good agreement between the CL-CIA method and the FAAS method, as demonstrated by a paired *t*-test (*t_stat_* = −1.09, and *t_crit_* = 2.78: *p* = 0.05). The variances in the results obtained using both methods were also not different (F*_stat_* = 0.32, and F*_cri_* = 5.32: *p* = 0.05). Six samples of paracetamol drugs, four in the form of tablets and two in the form of syrups, were employed to validate the CL-CIA system for the analysis of paracetamol. The samples were also analyzed using absorbance measurements at 243 nm with a UV–VIS spectrophotometer. The results in Figure 3b show that the values are comparable according to the paired *t*-test (*t_stat_* = 1.31, and *t_crit_* = 2.57: *p* = 0.05). The variances in the results are also not different (F*_stat_* = 0.02, and F*_cri_* = 4.96: *p* = 0.05).

### 2.6. Comparison with Other Chemiluminescence Flow Configurations

Table 2 compares six designs of CL flow cells, with descriptions of their features. All of these flow cells have more than one built-in inlet for the introduction of streams of samples and reagents to mix inside the flow cell. The ‘mixing chamber’ of design 1 in Table 2 [21] employs magnetic bar stirring to mix the samples and reagents, which are then dispensed into the 2000 µL chamber via the four built-in inlets. After detection with a photodiode placed above the chamber, the mixture is discarded as waste via the outlet located at the base of the chamber. Liquid flow uses the ‘flow-batch’ format, employing a peristaltic pump and five three-way solenoid valves. There is no information on how the photodiode is placed above the vessel to collect the CL emission or on the procedure used to record the light.

Designs 2 and 3 are flow cells in the shape of chambers, consisting of two parts, a cell body and a window, which form a closed flow cell. For the ‘vortex’ design (design 2), the cell body is made from a special ceramic material known as Green Tape™ 951 (DuPont, Wilmington, DE, USA) [22]. This cell is sealed with a glass sheet, acting as the detection window. Furthermore, to achieve vortex mixing in this flow cell, each inlet stream is split into two streams by machined grooves and holes placed within several ceramic layers in order to produce circular mixing in the chamber. This production requires special skills in designing and machining in order to exactly align the positions of the holes and the grooves of the individual layers prior to the thermocompression and sintering processes. For the cone-shape design (design 3) [23], the chamber body is made of Teflon with two inlets and one outlet. A transparent glass sheet is used to seal the cell, and it is employed as the CL window. The cell body also has a cross-flow point, where the streams of the sample and two reagents are merged before entering an inlet at the bottom of the flow cell. The cell has a second inlet positioned at the base for the introduction of a liquid carrier and an outlet positioned at the tip of the cone. The authors reported that this ‘cone-shaped’ design provides rapid homogeneity in less than 1 s, which is faster than the ‘mixing chamber’ of design 1 [21].

Flat flow cells are often used in chemiluminescence since their configurations are compatible with the flat geometry of PMTs. Table 2 lists (designs 4.1 to 4.3) a number of flat flow cells that are made using the CNC milling technique, namely, ‘spiral’, ‘sinusoidal’ and ‘serpentine’ cells. The spiral cell is designed similarly to coiled tubing in order to produce a horizontal spiral flow path compatible with the PMT window. For these three flat cells, the grooves are milled out on one side of a polymeric block in order to define the path of the solution flow. The grooves are then covered with a transparent material, such as epoxy acetate film [25] or a sapphire sheet [26], to form a channel. Each flow cell has two inlets and one outlet.

A flat spiral flow cell with built-in inlets and outlet can also be produced by using the 3D-printing technique. Design 4.4 was the first 3D-printed CL flow cell [24]. This spiral cell has the same design as the spiral cell produced via CNC milling [25,26]. The printed spiral cell is made from a UV-curable acrylic polymer. Post-printing processes are also used, including UV curing, to harden the polymer and to remove the supporting wax.

For the ‘lab-on-valve’ flow cell [39], after the first report of the integration of a Z-type multipurpose flow-through cell on the side of the LOV unit for chemiluminescence detection [36], two other studies of chemiluminescence measurements were carried out using the LOV flow system [27]. An interesting and effective approach was presented by Oliveira et al. [27], and this is designated as design 5 in Table 2. In their design, the PMT is mounted ‘atop’ of the LOV unit so that the light is collected in the reaction area. The system is covered with a black plastic box to prevent interference from external light.

The flow cell used in this work is also included in Table 2 as design 6. Compared with the other flow cells, the total internal volume of our cell (322 µL) is within the range (38–2000 µL) of that of the other reported flow cells. As shown in Table 2, the mixing chamber of design 1 has the highest volume of 2000 µL This rather large volume is not a problem, since it is equipped with a magnetic stirring system. It is estimated that the volume of the liquid required for the complete washing of this flow cell is 4.5 mL (3 × 1500 µL) per cycle. Design 1 is suitable for flow-batch systems but not for continuous flow systems. The other designs in Table 2, including our design, are suitable for continuous flow operations. The flow cells of designs 2 to 4 (including 4.1 to 4.4) all require window sealing. The CL detection in the LOV systems (designs 5 and 6) do not require window sealing. The LOV platform consists of four inlets (or more) and three outlets. In CIA, there are five inlets and five outlets built into the platform. Thus, the introduction and the draining of liquids in both the LOV and CIA systems are rapid. The cross-flow design of CIA provides turbulent mixing [37,38] inside the flow conduit, which is suitable for CL work. The detection of CL emissions in LOV and CIA systems employs a PMT placed over the platform. It can be concluded from Table 2 that the outstanding characteristic of our CIA platform is its simple fabrication. Further modification of the detection window is not necessary. Additionally, relatively small volumes of the sample and reagent solutions are required, making it cost effective, especially when using expensive reagents (e.g., luminol).

## 3. Materials and Methods

### 3.1. The CIA Platform Used for Chemiluminescence Detection

The CIA platform was produced similar to that described in the previous work [37]. In brief, the main flow path was produced by drilling a Perspex® block (50 mm × 30 mm × 15 mm) to make a straight channel (the x-axis channel). Four straight channels (y-axis channels) were drilled perpendicularly to this main channel across the width of the block. Sockets for screw nuts were made at both ends of each channel to connect the tubing (see Appendix A Appendix A).

The setup for the detection of CL included a light-tight box (130 mm × 70 mm × 30 mm) made from black cardboard, in which the CIA platform was placed. A circular hole (∅ = 5.5 cm) was drilled out at the top of the box to insert the lens of a Canon digital camera (EOS M50, Yokohama, Japan), as shown in Figure 1a, or to insert the entrance tube of the side-on photomultiplier tube (Oriel 7020 photomultiplier, Newport, Irvine, CA, USA), as shown in Figure 4a. The light-tight box also had 10 small holes (∅ = 2 mm), which were used to insert the tubing that was connected to the peristatic pumps. To prevent any stray light, the whole box was covered with a black cloth. The PMT had a detection area of ca.12.5 mm × 25 mm, covering the region of the No. 2 to No. 4 y-channels (see Appendix A). A DC power supply (C3830, Hamamatsu Corporation, Shizuoka, Japan) and a preamplifier unit (C7319, Hamamatsu Corporation, Shizuoka, Japan) were connected to the PMT. The signals from the PMT were recorded on a PC using Pico Log software (Pico Technology, Cambridge, UK).

### 3.2. CIA System I: Manifold and Operating Procedure for Measurement of Co (II)

Figure 4b presents a schematic diagram of the connection of the tubing to the CIA platform for the Co(II) solution and the CL reagents and the flow directions. The solutions of the Co(II) sample/standard (S) and the luminol (R1) and hydrogen peroxide reagents (R2 and R3) are pumped into the CIA platform via the y-axis channels denoted as No. 1 to No. 4, respectively. The x-axis channel is the H_2_O_2_ carrier (C). Appendix A shows a schematic of the complete flow system (System I), which is used for the analysis of Co(II). Tygon® pump tubing (0.95 mm i.d., Cole-Parmer, Vernon Hills, IL, USA) is used with the peristaltic pumps P1–P3 (Ismatec^TM^, Glattbrugg, Switzerland). PEEK tubing (1.0 mm i.d., VICI jour, Jonsered, Sweden) is used for all flow lines. The CIA System I is operated using an in-house control module, with the instructions written in Visual Basic 6.0 (see Appendix A in for the operating steps).

Briefly, at startup (Step 1), all the conduits are filled with their respective solutions, using pump P1 for the x-axis channel, pump P2 for y-axis channels 1 and 2, and pump P3 for y-axis channels 3 and 4 for 30 s. Then (Step 2), the cobalt (S) and luminol (R1) lines are stopped (pump P2) for 30 s, whilst the H_2_O_2_ solution in all 3 channels (C, 3 and 4) continuously flows. Next (Step 3), pump P2 is started to load fresh cobalt and luminol solutions into the platform, and pump P3 is stopped (the H_2_O_2_ reagent solution in y-axis channels) for 10 s. Then (Step 4), pump P2 is also stopped for 20 s. The H_2_O_2_ carrier solution flowing in the x-axis channel (C) then pushes and mixes the reagents from the 4 intersections in order to discard them as waste. Finally (Step 5), pump P3 is started in order to rinse out the x-axis channel with the H_2_O_2_ solution for 30 s.

For the subsequent sample/standard, Step 3 to Step 5 are repeated. It should be noted that the H_2_O_2_ carrier solution in the x-axis channel (C) flows continuously throughout the sequence of analyses.

### 3.3. CIA System II: Manifold and Operating Procedure for Measurement of Paracetamol

The CIA manifold, in terms of pumps, tubing connections and PMT, is the same as System I (see Appendix A for the full schematic details). Figure 4c shows the tubing connections used for the paracetamol analysis. Three reagents are used for the paracetamol analysis. These are Fe(CN)_6_^3−^ (R1), luminol (R2) and H_2_O_2_ (R3). The solutions of Fe(CN)_6_^3−^ (R1), the paracetamol sample/standard (S), luminol (R2) and the hydrogen peroxide reagent (R3) are pumped into the CIA platform via the y-axis channels denoted as No. 1 to No. 4, respectively. The x-axis channel is the H_2_O_2_ carrier (C). The operating steps are listed in Appendix A.

Briefly, at startup (Step 1), all the conduits are filled with their respective solutions, using pump P1 for the x-axis channel, pump P2 for y-axis channels 1–3 and pump P3 for y-axis channel 4 for 30 s. Then (Step 2), pump P2 is stopped for 30 s, whilst the H_2_O_2_ solution in channel 4 and the carrier line continuously flows. Next (Step 3), pump P2 is started in order to load the fresh sample and the 3 reagents into the platform, and pump P3 is stopped (the H_2_O_2_ reagent solution in channel 4 of y-axis) for 10 s. Then (Step 4), pump P2 is also stopped for 30 s. The H_2_O_2_ carrier solution flowing in the x-axis C channel then pushes and mixes the reagents from the 4 intersections in order to discard them as waste. Finally (Step 5), pump P3 is started in order to rinse out the x-axis C channel with H_2_O_2_ solution.

For the subsequent sample/standard, Step 3 to Step 5 are repeated. This is similar to the operation steps used in System I for the Co(II) analysis.

### 3.4. Standards and Reagents

All chemicals are of analytical reagent grade. Deionized-distilled Milli-Q^®^ water (18 MΩ · cm) is used throughout. The standard and reagent solutions employed in the CIA systems for the analysis of Co(II) and paracetamol are prepared as described for each procedure (vide infra). 

#### 3.4.1. Analysis of Co (II)

A stock standard solution of cobalt (II) (1.0 mol L^−1^) is prepared by dissolving 7.5 g of cobalt nitrate hexahydrate (Ajax Chemicals, Hindmarsh, Australia) in 1% *w*/*v* nitric acid (Sigma-Aldrich, Taufkirchen, Germany) to a final volume of 25.00 mL. The accurate concentration of this stock solution is determined using flame atomic absorption spectrometry (FAAS), calibrated with a reference solution of Co(II) solution for AAS (Sigma-Aldrich, Taufkirchen, Germany). A series of working standard Co(II) solutions (0.002–0.025 mg L^−1^) are prepared from the stock standard solution via an appropriate dilution with water.

The 2.5 mmol L^−1^ luminol solution (R1 in Figure 4b) is freshly prepared by dissolving approximately 50 mg of luminol powder (Sigma-Aldrich, Taufkirchen, Germany) in 100 mL of 0.05 mol L^−1^ NaOH (Merck, Darmstadt, Germany). The luminol reagent is stored in a brown bottle. The H_2_O_2_ reagent (R2) and carrier (C) solution (Figure 4b) is a 0.5% *w*/*w* H_2_O_2_ solution, prepared via an appropriate dilution of 30% *w*/*w* H_2_O_2_ (Merck, Darmstadt, Germany) with 0.1 mol L^−1^ carbonate buffer (pH 10.5).

#### 3.4.2. Analysis of Paracetamol

A stock standard solution of paracetamol (1000 mg L^−1^) is prepared by dissolving an accurate weight of 0.100 g of pure paracetamol (Sigma-Aldrich, Taufkirchen, Germany) in 100.0 mL of water. The working standard solutions (10 to 200 mg L^−1^) are prepared via an appropriate dilution with water from the stock solution. Stock ferricyanide reagent is prepared by dissolving 3.3 g of K_3_Fe(CN)_6_ crystal (Merck, Darmstadt, Germany) in 100 mL of 0.05 mol L^−1^ phosphate buffer solution (pH 11.5). Reagent R1 (Figure 4c) is prepared from this stock reagent via dilution with the same buffer. The luminol reagent (R2 in Figure 4c) is prepared by dissolving 0.06 g of luminol powder (Sigma-Aldrich, Taufkirchen, Germany) in 100 mL of the 0.05 mol L^−1^ phosphate buffer. The H_2_O_2_ reagent (R4) and carrier (C) solution (1% *w*/*w*) is prepared by diluting 33.3 mL of 30% *w*/*w* hydrogen peroxide with 0.05 mol L^−1^ phosphate buffer solution (pH 11.5) to 1000 mL.

### 3.5. Sample Preparation

Four kinds of water samples were employed in the validation study of the CL-CIA system for Co(II) determination. Deionized (DI) water was obtained from the Milli-Q water purifying system. Bottled drinking water was purchased from a local supermarket in Bangkok. Tap water was collected from the water supply of the faculty. Canal water samples (A and B) were collected from two sites in Bangkok. The canal water samples were filtered through a 0.2 µm syringe filter prior to injection into the CIA System I (Figure 4b), whereas the other water samples were analyzed directly. The quantitative analysis of the paracetamol employed six pharmaceutical drugs (4 tablet and 2 syrup samples). The samples were purchased from local drug stores in Bangkok. For the tablet form of paracetamol, the sample was ground in a mortar. An accurate weight of 0.100 g was dissolved in 10.00 mL of DI water. The solution was filtered through a 0.45 µm syringe filter prior to being pumped into the CIA platform (Figure 4c). The syrup samples were diluted with DI water to provide concentrations within the linear range of the method.

## 4. Conclusions

This is the first report using CIA for chemiluminescence detection. The transparent property of the acrylic material allows for the detection of CL light directly from the flow manifold. The CIA platform consists of a small acrylic block (ca. matchbox size) with five straight cylindrical channels, with four of the channels set perpendicularly to the main flow-channel, which is set along the length of the device. The sample and reagents are introduced into the platform along these four cross-flow channels. The flowing carrier solution in the main flow channel mixes the sample and reagent(s) contained in the volumes of the four intersecting channels. The platform acts as both a CL reactor and a CL flow cell. A side-on PMT is placed above the CIA platform for the in situ collection of the CL emission along the main flow channel. The CL-CIA system was applied to the analysis of Co(II) in water using the Co(II)-catalyzed H_2_O_2_-luminol reaction. In a converse manner, the reduction of the CL of the Fe(CN)_6_^3−^-catalyzed H_2_O_2_-luminol via the reaction between paracetamol and Fe(CN)_6_^3−^ was employed for the determination of paracetamol. Compared with the other CL flow cells with two or more inlets (Table 2), our cell volume (322 µL) is comparable to the average cell volume of these systems (332 µL, *n* = 13). The CL-CIA platform has five inlets and five outlets, which allow for the rapid loading and dispensing of solutions.

## Figures and Tables

**Figure 1 molecules-28-01316-f001:**
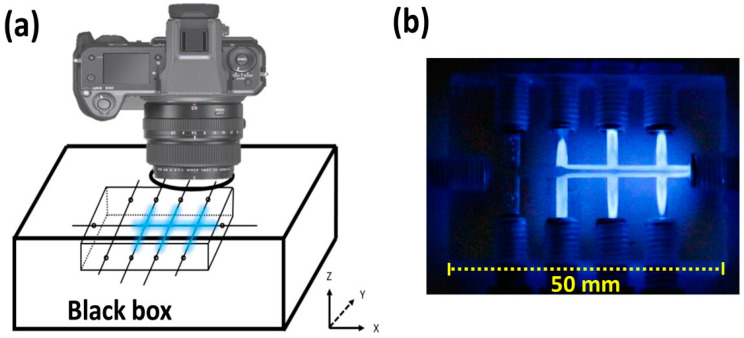
Schematics of the CL-CIA platform in preliminary experiments showing (**a**) camera position on top of the light-tight box and (**b**) an example of the recorded image of the platform conduit of the blue light emitted from the mixing of the chemiluminescent reagents for Co(II)-catalyzed H_2_O_2_-luminol reaction. The concentration of standard solution of Co(II) is 10 mmol L^−1^. Note: the CIA manifold is shown in more detail in Appendix A.

**Figure 2 molecules-28-01316-f002:**
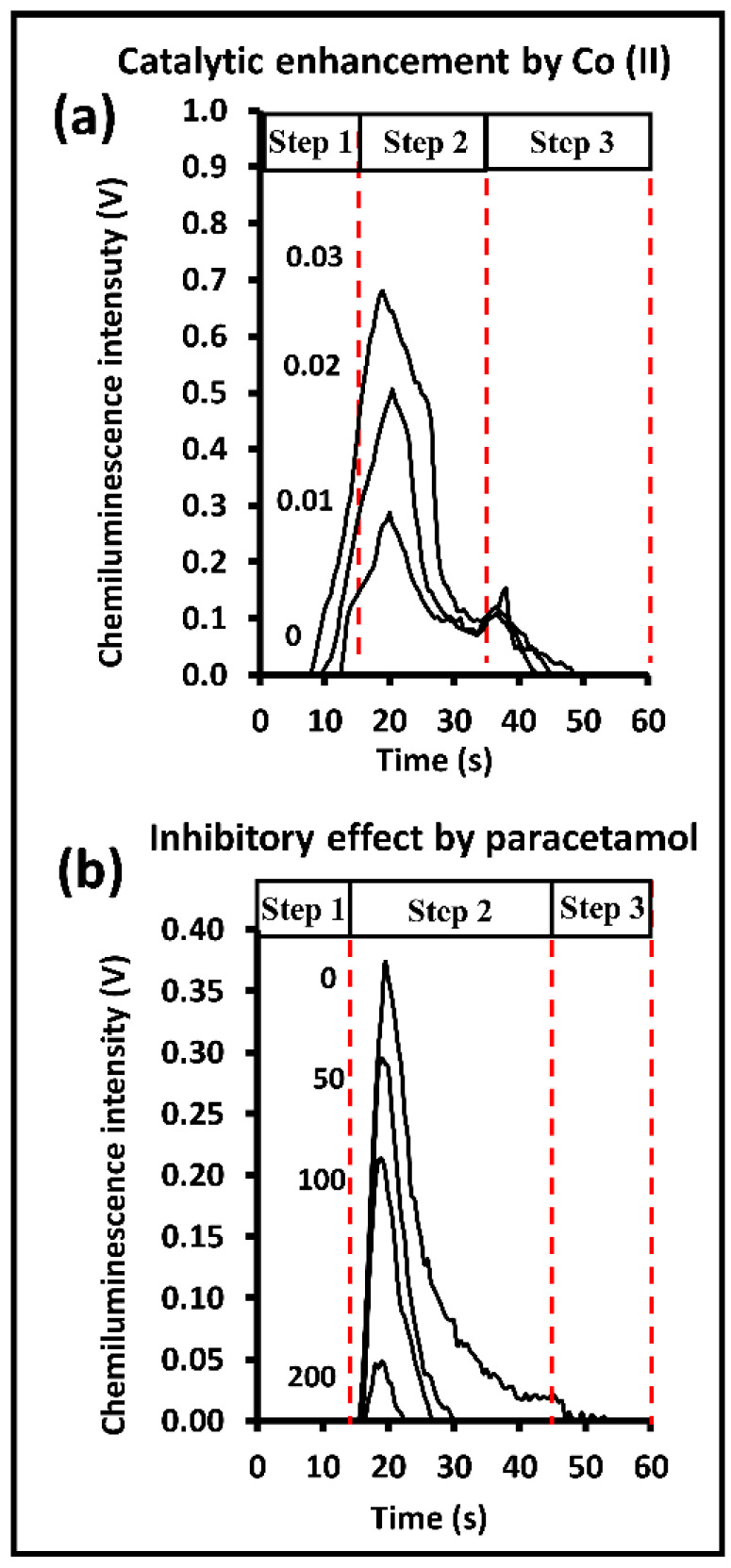
Examples of chemiluminescence signal profiles obtained from the CIA flow manifolds in Appendix A. (**a**) Catalytic enhancement of CL emission induced by three concentrations of standard solutions of Co (II) (0.01–0.03 mg L^−1^). (**b**) Inhibitory effect induced by three concentrations of standard solutions of paracetamol (50–200 mg L^−1^).

**Figure 3 molecules-28-01316-f003:**
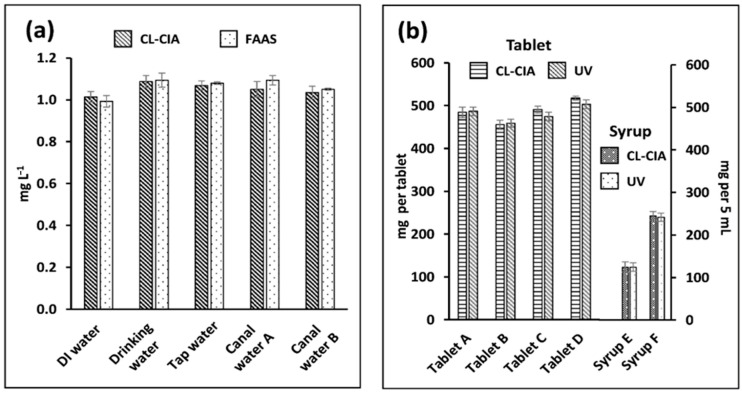
Results of the CL-CIA methods for the quantification of (**a**) Co(II) spiked in water samples and (**b**) paracetamol (tablets and syrups). The results are compared with the reference methods, i.e., FAAS for Co(II) and UV absorbance at 243 nm for paracetamol.

**Figure 4 molecules-28-01316-f004:**
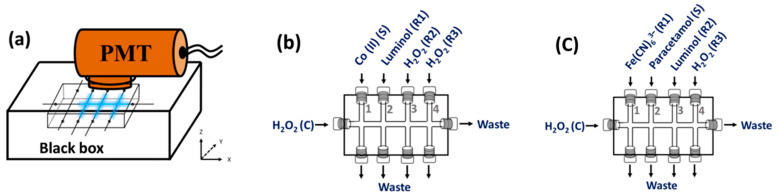
Schematics of the setup of the CIA platform: (**a**) area (blue zone) of collection of light by the side-on PMT, (**b**) flow arrangement for Co(II) analysis and (**c**) flow arrangement for paracetamol analysis. See Appendix A for details of the final setup of the CIA-CL systems for Co(II) and paracetamol analyses, respectively.

**Table 1 molecules-28-01316-t001:** Analytical features of the CL-CIA systems for Co(II) and paracetamol.

Feature	Cobalt (II) Analysis	Paracetamol Analysis
Working range	0.002–0.025 mg L^−1^	10.0–200 mg L^−1^
Calibration equation	Signal (volt) = (69.3 ± 2.34) · (Co(II), mg L^−1^) + (2.50 ± 0.03), r^2^ = 0.9977	% decrease = (0.34 ± 0.02) · (paracetamol, mg L^−1^) +(6 ± 2), r^2^ = 0.9906
Limit of quantitation	0.0016 mg L^−1^(2 SD of y-intercept/slope)	3.87 mg L^−1^(3SD of blank/slope)
Precision (RSD)	1.9–3.7% (*n* = 30)	0.3–4.2% (*n* = 25)
Recovery	87–103%	97–110%
Sample throughput	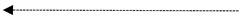 60 samples h^−1^ 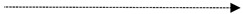
Sample and reagent consumption	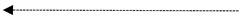 167 µL 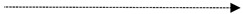

**Table 2 molecules-28-01316-t002:** Comparison of chemiluminescence flow cells with built-in ports for sample or reagent(s) introduction and an outlet on the cell body.

Design	FabricationTechnique	Body Material/Cell Volume (µL)	Window Material	Remark
1. Mixing chamber (4 inlets and 1 outlet) [21]	Not mentioned	Teflon block/2000	Not mentioned	Cylindrical chamber for dispensing sample and reagents via stirring systemPhotodiode placed above the chamber
2. Vortex (4 inlets and 1 outlet) [22]	Thermocompressionand sintering	Fused multilayer ceramic/38, 152, 271	Glass sheet	Requires special ceramic materials and specific design and production skills
3. Cone-shaped(2 inlets with a cross-flow point and 1 outlet) [23]	Not mentioned	Teflon block/280	Glass sheet	Three liquids (i.e., sample, reagent 1 and reagent 2) can be pre-mixed at the cross-flow point before entering the cone cell
4. Flat flow cells with channels				
4.1. 3D-printed spiral (2 inlets and 1 outlet) [24]	3D printing	Visijet Ex200 UV curable acrylic plastic as build material/144	Epoxy acetate film	Employs 3D printer with use of wax (Visijet S200) as the support material and post-printing processes
4.2. Milled spiral (2 inlets and 1 outlet) [25,26]	CNC milling	Poly carbonate/144 [25]White acetal/144 [25]Teflon disk/275 [26]	Epoxy acetate film [25]Sapphire window [26].	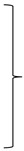	Computerized controlled milling machine is necessaryEasy to clean the channel before sealing with window
4.3. Milled sinusoidal (2 inlets and 1 outlet) [25]	CNC milling	White acetal block/133	Epoxy acetate film
4.4. Milled serpentine (2 inlets and 1 outlet) [25,26]	CNC milling	White acetal/144 [25]Teflon disk/275 [26]	Epoxy acetate film [25]Sapphire window [26]
5. Lab-on-valve flow unit (4 inlets or more and 3 outlets) [27]	Mesofabrication [39]	Perspex block/Not mentioned	Not needed	PMT covers the reaction area on platform
6. Cross-flow platform (5 inlets and 5 outlets) (used in this work)	Milling and drilling	Perspex block/322	Not needed	PMT covers the reaction area on platformDue to simple design with straight channel conduit, manual milling and drilling machines can be used

## Data Availability

Not applicable.

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
