# Peer review of "Transparent Cross-Flow Platform as Chemiluminescence Detection Cell in Cross Injection Analysis"

_molecules, 2023, doi:10.3390/molecules28031316_

Round 1

Reviewer 1 Report

Cross Injection Analysis was introduced in 2013 by the same group of researchers as an alternative approach to perform flow analysis with simplified instrumentation (only peristaltic pumps are employed). SInce the first introduction of CIA only 2 additional articles have been published also by the same group in 2015 and 2016. 

In the present submission the authors take advantage of the optical properties of the material used for the construction of the CIA manifold that can serve as a flow cell for CL analyses. In brief the authors in a rather expected way placed the manifold in a black box that houses a PMT at a suitable position oposite to the mixing tees of the manifold. Two well known CL chemistries were selected as a proof of concept.  

I am sceptical about the novelty of this work both from instrumental and certainly from an application point of view. What the authors did instrumentaly is more or less a straightforward approach, at least to me. Besides, CIA does not seem to have attracted much audience since its introduction. 

I am therefore not able to propose acceptance of this article in Molecules as a full research article. If novel CL chemistries have been proposed by the authors my opinion might be different. I could however consider a "communication" following proper modification of the article (reduction etc) 

Reviewer 2 Report

Comments and suggestions for Authors:

-          Section 2.5. In addition to the t-test, the authors should also perform the F-test of equality of variances.

-          Lines 288-289. The sentence ‘The LOV platform consists of 4 outlets (or more) and 3 outlets’ is not clear.

-          Figure S4 (in Supplementary information). It is not obvious where evaluation areas A and B are located. If these are rectangles marked with a dashed line in the photos shown in Figure S4 (a-d), this should be clearly indicated.

Reviewer 3 Report

This manuscript designed a transparent cross-flow platform as a chemiluminescence detection cell for cross injection analysis (CIA), which can be applied to the analysis of Co(II) and paracetamol. It is the first report of CIA for chemiluminescence detection. However, the innovation and significance of this work cannot be fully understood through the description and elaboration of the author. Therefore, this article is not recommended to be published in Molecules.

1. The introduction section only introduces the basic research status, without analyzing the existing problems of this field nor the significance of this work, which makes it impossible to understand why the author did this work, and what innovation does the work have compared with the previous reports.

2. All the expressions are very difficult to understand. The language of the manuscript should be improved.

3. A length scale should be added in Figure 1b to make the actual size of the device more clear.

4. It seems like that the linear calibrations of Co(II) and paracetamol are the key data of this work which have been repeatedly mentioned in the abstract and the whole manuscript. But there is even no direct data graph to demonstrate the results of linear calibration. The author should make corresponding supplements.

5. What’s the meaning of Table 2? What conclusions can be drawn from this comparison?

Round 2

Reviewer 1 Report

The revised version could be accepted for publication